# Identification of Adipose Tissue as a Reservoir of Macrophages after Acute Myocardial Infarction

**DOI:** 10.3390/ijms231810498

**Published:** 2022-09-10

**Authors:** Ingrid Gomez, Virginie Robert, Paul Alayrac, Adèle Arlat, Vincent Duval, Marie-Laure Renoud, José Vilar, Mathilde Lemitre, Jean-Sébastien Silvestre, Béatrice Cousin

**Affiliations:** 1PARCC, INSERM, Université Paris Cité, 75015 Paris, France; 2RESTORE Research Center, Université de Toulouse, INSERM 1301, CNRS 5070, EFS, ENVT, 31100 Toulouse, France

**Keywords:** macrophages, myocardial infarction, adipose tissue, diabetes

## Abstract

Medullary and extra-medullary hematopoiesis has been shown to govern inflammatory cell infiltration and subsequently cardiac remodeling and function after acute myocardial infarction (MI). Emerging evidence positions adipose tissue (AT) as an alternative source of immune cell production. We, therefore, hypothesized that AT could act as a reservoir of inflammatory cells that participate in cardiac homeostasis after MI. To reveal the distinct role of inflammatory cells derived from AT or bone marrow (BM), chimeric mice were generated using standard repopulation assays. We showed that AMI increased the number of AT-derived macrophages in the cardiac tissue. These macrophages exhibit pro-inflammatory characteristics and their specific depletion improved cardiac function as well as decreased infarct size and interstitial fibrosis. We then reasoned that the alteration of AT-immune compartment in type 2 diabetes could, thus, contribute to defects in cardiac remodeling. However, in these conditions, myeloid cells recruited in the infarcted heart mainly originate from the BM, and AT was no longer used as a myeloid cell reservoir. Altogether, we showed here that a subpopulation of cardiac inflammatory macrophages emerges from myeloid cells of AT origin and plays a detrimental role in cardiac remodeling and function after MI. Diabetes abrogates the ability of AT-derived myeloid cells to populate the infarcted heart.

## 1. Introduction

Following injury, damaged tissue is repaired through the coordinated biological actions of different cell types, among which immune cells play a critical role. In a steady state condition, leukocytes mainly derive from bone marrow (BM) hematopoietic stem cells (HSCs). Remarkably, recent works have challenged the view of the medullary origin of hematopoiesis and found that systemic inflammation can also stimulate extramedullary hematopoiesis in adult mice and humans. Consistent with this, emerging evidence positions adipose tissue (AT) as an alternative reservoir of inflammatory cell production. Indeed, a significant population of cells harboring the same phenotype as BM-derived HSCs have been identified in AT [1,2]. These cells exhibit the functional features of bona-fide HSCs, such as multipotency and self-renewal. In addition, AT-derived HSCs present an intrinsic potential to replenish all major hematopoietic lineages in long term reconstitution assays, as well as to efficiently generate myeloid cells within AT but not in conventional hematopoietic organs, suggesting a priority homing to AT [2]. This endogenous AT hematopoietic activity contributes to tissue homeostasis not only in the AT-immune compartment, but also in other organs [3]. Indeed, AT could serve as a reservoir of myeloid cells able to contribute to tissue repair processes in the AT itself, and also in other organs. For example, AT-derived macrophages are required for AT regeneration [4] and AT-derived mast cells are capable of homing to different organs such as the skin and gut, where they acquire the properties of functional tissue mast cells [5]. In agreement with this, cardiac mast cells populating the ischemic milieu mainly arise from AT [6]. Such endogenous AT hematopoiesis is altered in diabetes, leading to the production of inflammatory cells that sustain chronic low-grade inflammation [7], suggesting that AT reservoir potential and function could be modified in metabolic disease and could contribute to alteration of tissue repair process in the diabetic setting.

In the heart, both the innate and adaptive immune systems have been shown to regulate cardiac remodeling and function after acute myocardial infarction (MI). In particular, monocytes and macrophages exhibit a central role in the acute and chronic phases of post-MI cardiac repair, as the balance between their pathogenic and pro-healing function controls the disease outcome [8]. Of note, Ly6C^high^ monocytes are continuously recruited into ischemic tissue in the first week after MI, and maintenance of this influx depends on monocyte mobilization from a splenic reservoir [9], as well as activation of emergency myelopoiesis to increase monocyte production in the bone marrow (medullary monopoiesis) and the spleen (extramedullary monopoiesis) [10,11]. Many of these monocytes may either die or exit the cardiac tissue, whereas surviving monocytes populating the ischemic milieu may acquire distinct macrophage phenotypes associated with specific functions in the resolution of inflammation, tissue repair and remodeling [12]. Consistent with this, macrophages mediate the efferocytosis of injured cardiomyocytes, favoring the secretion of the proangiogenic and antifibrotic vascular endothelial growth factor to locally repair the dysfunctional heart [13].

Of great interest, in the setting of type 2 diabetes, monocyte and macrophage fates switch toward an inflammatory phenotype that precipitates adverse ventricular remodeling after acute MI [14]. Studied diabetic patients displayed an imbalanced inflammatory versus reparative macrophage ratio attributable to a reduction in the number of reparative cells [15]. The inflammatory versus reparative macrophage ratio was also directly correlated with waist circumference and glycated hemoglobin. In a rat model of type 2 diabetes, the heme oxygenase inducer, hemin, promoted preferential polarization of macrophages toward anti-inflammatory phenotype in cardiac tissue with concomitant improvement of cardiac function [16]. Diabetic mice had increased numbers of circulating Ly6C^high^ monocytes, reflecting hyperglycemia-induced proliferation and expansion of bone marrow myeloid progenitors, as well as release of monocytes into the circulation. Remarkably, treatment of hyperglycemia reduced monocytosis [17]. Hence, diabetes could also shape inflammatory cell infiltration through activation of monocyte mobilization from medullary and extra-medullary reservoirs, including AT.

We, therefore, hypothesize that, in addition to metabolic and endocrine signals, AT could act as a reservoir of myeloid cells that participate in cardiac remodeling after MI. We also speculate that diabetes could impact inflammation-dependent cardiac remodeling through the activation of AT hematopoiesis.

## 2. Results

### 2.1. Establishment of Experimental Models

To reveal the distinct role of macrophages derived from AT or BM, chimeric mice were generated using standard repopulation assays, as previously described [2]. Briefly, lethally irradiated wild-type C57Bl/6 mice were reconstituted with 2.10^3^ AT-HSC (AT-mice) or BM-HSC (BM-mice) sorted from tdTomato mice and with 2.10^5^ BM cells isolated from wild-type C57Bl/6 mice (Figure 1A). Injection of BM cells ensured the survival of both AT-mice and BM-mice, as previously described [2,18]. Chimerism was determined by flow cytometry (Figure 1B) in AT, BM and spleen. Total chimerism in the AT reached 40 to 70% in AT- and BM-chimeric mice, respectively (Figure 1C). In contrast, chimerism in hematopoietic organs, i.e., BM and spleen, was detected only in BM-chimeric mice and not in AT-chimeric animals (Figure 1C), as previously described [2]. Using these experimental models, CD45^+^/dT^+^ cells were identified, and represented around 3–4% of total immune cells in the heart of AT-mice, suggesting that a small population of CD45^+^ cells originating from AT populated the cardiac tissue in steady state conditions (Figure 1C).

To assess the in vivo functional relevance of our experimental models, chimeric mice were subjected to myocardial infarction (MI), 8 weeks after reconstitution. Cardiac function was evaluated before and after MI in AT- and BM-chimeric mice. The left ventricle ejection fraction was similar in AT- and BM-mice before surgery (Figure 1D). As expected, MI prompted a severe decrease in left ventricle ejection fraction and a significant increase in infarct size as well as interstitial fibrosis, at 2 weeks after the onset of ischemia (Figure 1E). MI related effects on cardiac function and remodeling were similar in AT- and BM-mice, validating the use of these experimental models (Figure 1E).

### 2.2. A Subset of Cardiac Inflammatory Myeloid Cells Emerges from AT-Endogenous Hematopoiesis

We next hypothesized that MI could prompt the recruitment and infiltration of inflammatory cells originating from AT hematopoiesis within the cardiac tissue. Flow cytometry analyses were, thus, performed to identify dT^+^ immune cells in the heart and blood of AT-chimeric mice (Figure 2 and Appendix A). No CD45^+^dT^+^ cells were present in the blood of mice before surgery, and only a few were identified in the heart (Figure 2A). MI increased CD45^+^dT^+^ cell numbers and percentage in blood and heart, reaching a maximum on day 7 after MI (Figure 2A). We then attempted to characterize the type of cardiac CD45^+^dT^+^ cells in this experimental setting at 7 days post-MI. dT^+^ cells were identified as mainly myeloid cells of monocyte and macrophage phenotype. dT^+^ cells were not detected within the population of lymphoid cells including T and B lymphocytes (Figure 2B).

To further confirm the relevance of the AT reservoir, we compared the number of monocytes and macrophages in both AT-mice and BM-mice. We showed that AT- and BM-mice displayed similar numbers of total and dT^+^ cardiac monocytes and macrophages at day 7 after MI (Figure 2C). Equivalent amounts of dT^+^ monocytes were also found in blood, BM and spleen, 7 days post-MI (Appendix AA) showing that the engraftment of AT-derived myeloid cells was similar to medullar cells. To compare cells of adipose and BM origin, dT^+^ macrophages were sorted from the hearts of AT- and BM-chimeric mice, 7 days post-MI. dT^+^ macrophages from AT-mice and from BM-mice displayed similar expression of inflammatory cytokines such as IL-1β and IL-6 (Figure 2D), as well as of IFNγ, TNFα, IL-12p40, and chemokines (CCL2, CCL3 and CCL11) (Appendix AB). In contrast, the production of anti-inflammatory cytokine IL-10 was significantly lower in cardiac macrophages sorted from AT-mice compared with BM-mice (Figure 2D), suggesting that cardiac macrophages derived from AT-endogenous hematopoiesis likely exert pro-inflammatory function.

These results demonstrate that pro-inflammatory myeloid cells originating from endogenous AT hematopoiesis are able to transiently infiltrate the heart after MI, where they are expected to exhibit pro-inflammatory characteristics and subsequently have deleterious effects on cardiac homeostasis.

### 2.3. Specific Depletion of AT-Derived Myeloid Cells Improves Cardiac Remodeling after MI

To determine the role of AT-derived macrophages in cardiac remodeling after MI, a hematopoietic repopulation assay was performed, using AT-HSC or BM-HSC isolated from mice expressing the diphtheria toxin receptor (DTR) under control of the full CD11c promoter [7], mixed with competitor BM cells from mT/mG mice, and transplanted into lethally irradiated mT/mG recipient mice. Chimeric mice received either diphtheria toxin (DT) in order to specifically deplete CD11c^+^ macrophages deriving from AT or BM, or PBS (Figure 3A). In both AT- and BM-mice, DT treatment reduced cardiac CD11c^+^ cells, and more specifically, cardiac CD11c^+^ macrophages (Figure 3B). As a result, cardiac function was improved as revealed by the increase in left ventricle ejection fraction in DT-treated mice compared with controls (Figure 3C). In addition, whereas capillary density was unaffected, both infarct size and interstitial fibrosis were significantly decreased after DT treatment (Figure 3C).

Finally, DT treatment reduced pro-inflammatory cytokines such as IL-1β, IL-6, IFNγ, and TNFα in the heart (Figure 4A) and blood (Figure 4B) in both AT- and BM-mice. DT treatment also decreased chemokines such as CCL2, CCL3, and CCL4 in the heart, confirming the local inflammation decline (Appendix AA,B). These results suggest that pro-inflammatory CD11c^+^ cell-derived macrophages from AT origin display deleterious effects in the infarcted heart.

### 2.4. Diabetes Abrogates AT-Derived Cell Supply in the Heart

We then reasoned that pathological conditions affecting AT, such as a high fat diet (HFD), may exacerbate the contribution of AT-derived macrophages within cardiac tissue. For this purpose, AT- and BM-chimeric mice were fed either with normal chow (NC) or a HFD for 8 weeks (Figure 5A). HFD has previously been described to induce insulin resistance and dysglycemia in mice, including chimeras [7,19]. We first characterized the metabolic profile of these chimeric mice. As expected, after 8 weeks of HFD, total body weight and sub-cutaneous AT weight were slightly increased (Appendix AA). Chimeric mice became glucose intolerant as revealed by an intraperitoneal glucose tolerance test, and their fasting glucose levels were significantly higher than those of normal chow fed mice (Appendix AB). In MI mice, HFD induced a decrease in left ventricle ejection fraction (Figure 5B), and an increase in infarct size and interstitial fibrosis (Figure 5C), confirming that HFD worsens post-ischemic cardiac function and remodeling.

This alteration in cardiac remodeling was associated with an increase in Ly6C^High^ and Ly6C^Low^ monocytes and macrophages in the heart of HFD mice compared with NC animals, 7 days post-MI in both AT- and BM-chimeric mice (Figure 6A). However, in AT-chimeric mice, the percentage of dT^+^ cells within the myeloid populations did not increase in parallel (Figure 6A) suggesting that, in diabetic mice, myeloid cells recruited in the infarcted heart originated from the BM, and that AT was no longer used as a myeloid cell reservoir. The accumulation of cardiac monocytes and macrophages was associated with an increase in pro-inflammatory cytokines (IL-1β, IL-6, IFNγ) in the heart (Figure 6B) and blood (Figure 6C), suggesting a pro-inflammatory polarization for these macrophages. Chemokines levels were not changed in diabetic mice in both the heart (Appendix AC) and blood (Appendix AD), except for CCL4 that was increased in diabetic conditions in the infarcted heart.

To confirm that macrophages deriving from endogenous AT hematopoiesis did not play a role in cardiac remodeling after MI under diabetic conditions, CD11c^+^ macrophages stemming from AT-hematopoiesis were specifically depleted using the strategy described above. Briefly, irradiated mice were reconstituted with AT-HSC or BM-HSC cells sorted from CD11c-DTR mice, before being fed with an HFD or an NC for 8 weeks. DT treatment was achieved before MI, and cardiac function and inflammation were estimated 14 days post-surgery. In diabetic BM-chimeric mice, DT treatment improved cardiac function and remodeling, as revealed by the increase in left ventricle ejection fraction and capillary density, and the decrease in interstitial fibrosis and infarct size (Figure 7A). Nevertheless, DT treatment did not restore cardiac function and remodeling in treated diabetic AT-chimeric mice (Figure 7A). In this line of reasoning, DT treatment decreased monocyte and macrophage numbers (Figure 7B) as well as pro-inflammatory cytokine levels (IL-1β, IL-6 and IFNγ) (Figure 7C) in diabetic BM-chimeric mice but not in diabetic AT-chimeric mice. Altogether, these results suggest that diabetes abrogates the ability of AT-derived myeloid cells to populate the infarct heart.

## 3. Discussion

In the present work, we demonstrated that a subset of cardiac inflammatory macrophages emerged from myeloid cells of AT origin and played a deleterious role in post MI cardiac remodeling. Furthermore, in diabetic conditions that worsen post-ischemic cardiac function and remodeling, the AT was no longer used as a reservoir for inflammatory macrophages.

Cardiac macrophages are a heterogeneous cell population of distinct origin exhibiting specific phenotype, function and localization. In steady state conditions, the adult heart contains subsets of resident macrophages of embryonic origin [20]. The low number of AT-derived cells in physiological conditions likely precludes a substantial role for these cells in cardiac homeostasis. In this line of reasoning, in physiological conditions, AT-derived immune cells primarily repopulate the AT immune compartment [2]. In sharp contrast, the contribution of the AT reservoir to the pool of cardiac monocytes and macrophages is substantial after acute MI. Ly6C^High^ monocytes are continuously recruited into the ischemic tissue in the first week after MI, and maintenance of this influx depends on monocyte mobilization from a splenic reservoir and activation of emergency myelopoiesis to increase monocyte production in the bone marrow (medullary monopoiesis) and in the spleen (extramedullary monopoiesis) [10,11]. One can then speculate that AT could participate in the activation of emergency extramedullary myelopoiesis to increase inflammatory cell production and govern cardiac repair after acute MI.

Our results indicated that macrophages derived from AT-HSC promoted adverse ventricular remodeling and cardiac dysfunction. The deleterious role of AT-derived macrophages in post-ischemic cardiac remodeling contrasts with the beneficial function of these cells in the control of AT homeostasis and repair [4]. This dual role emphasizes the importance of macrophage imprinting by the cellular niche they reside in. Indeed, it has been shown that circulating monocytes engrafting in tissues, acquire the phenotype and functional properties of their resident counterparts [21]. Early after MI, the cardiac microenvironment is filled with pro-inflammatory mediators that are able to drive the polarization of macrophages [22]. One can, thus, postulate that AT-derived macrophages are shaped by the inflammatory post-ischemic cardiac niche and, thus, adopt an inflammatory profile, contributing to adverse remodeling of the infarcted heart.

This also indicates that cardiac macrophage states are strongly influenced by their tissue microenvironment, likely leading to substantial differences between mice and human macrophage-related effects. Macrophages within the mouse heart can be segregated into CCR2^−^ and CCR2^+^ subsets with distinct origins, repopulation processes and functions. On the same note, the human cardiac tissue comprises CCR2^−^ and CCR2^+^ macrophages with distinct functional properties and origins, equivalent to that of murine macrophages [23]. Nevertheless, even by using fate mapping studies in mice and single-cell RNA sequencing in humans to identify the most transcriptionally conserved subset of macrophages, these cells still display species-specific transcriptional differences [24]. In addition, human macrophages are a highly heterogeneous population of cells. It has been shown that stimulation of human macrophages with diverse activation signals, leads to the acquisition of a data set of 299 macrophage transcriptomes. Hence, although central transcriptional regulators associated with all human macrophage activation have been identified, they are under the control of stimulus-specific programs [25]. Furthermore, spatial multi-omic mapping of human myocardial infarction also reveals that a given cell state changes based on the cells’ neighborhood. For example, gene-regulatory programs driving injury of cardiomyocytes, activated phagocytic macrophages and their relation to myofibroblast differentiation in cardiac tissue remodeling are dependent on specific myocardial tissue zones and disease stages [26]. Hence, tissue macrophage heterogeneity is a critical determinant of immune responses and likely governs the distinct ability of human or mouse macrophages to shape the infarcted heart.

Diabetes is associated with an increase in total macrophage number and pro-inflammatory cytokine production in the infarcted myocardium. In agreement with these data, we have previously shown that diabetes induces an alteration of endogenous myelopoiesis in the AT that leads to the production of pro-inflammatory macrophages [7]. A similar mechanism has been described for medullar HSC that undergo an epigenetic modification in diabetic mice, leading to increased macrophage number and pro-inflammatory polarization [27,28]. Although the total number of macrophages increased in the infarcted myocardium, the percentage of myeloid cells arising from the AT was severely decreased, showing that in diabetic conditions cardiac macrophages preferentially derive from circulating medullar monocytes. In agreement with this, the specific depletion of AT-derived macrophages had no effect on cardiac function after MI, whereas improvement of cardiac function was observed following depletion of BM-derived macrophages. Hence, in diabetes, AT-derived macrophages do not engraft in the myocardium and are, thus, not involved in cardiac remodeling. It is likely that medullar macrophages have a competitive advantage over AT-derived macrophages for engraftment in the diabetic cardiac niche. In this line of reasoning, the diabetic heart is characterized by structural and metabolic abnormalities [29,30] that could explain differential cell engraftment.

We identified a novel population of cardiac macrophages that derives from endogenous AT hematopoiesis and accumulates in the infarcted myocardium, harboring deleterious effect in cardiac homeostasis. Identification of the molecular and cellular mechanisms that control AT-derived macrophages recruitment and polarization will provide new insights in the regulation of cardiac remodeling following MI.

## 4. Materials and Methods

*Animals:* Experiments were performed on 5 to 7 week-old male C57BL/6 mice (Envigo, Gannat, France), congenic male B6.129(Cg)-Gt(ROSA)26Sortm4(ACTB-tdTomato,-EGFP)Luo/J (mTmG mice, The Jackson Laboratory, Bar Harbor, USA, stock N°007676), and CD11c^+^-DTR mice. Animals were group-housed in a controlled environment (12 h light/dark cycles at 21 °C) with unrestricted access to water and a standard chow diet in a pathogen-free animal facility. For CD11c^+^ cell depletion, mice received an intraperitoneal injection of diphtheria toxin (20 ng/kg) at day 0, and 3 and 6 days after the onset of MI. Mice were euthanized by cervical dislocation under prior sedation by inhalation of isoflurane (2.5%).

*Single cell suspensions*: At mice necropsy, BM cells were immediately flushed from the sampled femurs and tibias with α-MEM medium (Life Technologies, Saint Aubin, France). Splenocyte suspensions were obtained by crushing the spleen through a 40 μm cell strainer and a red blood cell lysing buffer (StemCell Technologies, Saint Egrève, France) was used to remove red blood cells. Subcutaneous AT was carefully dissected, mechanically dissociated and digested at 37 °C with collagenase (Roche Diagnostics, Mannheim, Germany) for 30 min. Cells from the AT stroma vascular fraction (SVF) were collected by centrifugation after elimination of undigested fragments by filtration, as previously described [2]. Red blood cells were removed by incubation in hemolysis buffer (140 mM NH4Cl and 20 mM Tris, pH 7.6). Left ventricles were collected and minced with fine scissors, gently passed with PBS/1% fetal bovine serum (FBS) through a 12-well Tissue Disaggregator (Bel-ArtTM SciencewareTM, Fisher Scientific, Illkirch-Graffenstaden, France), and then filtered through 40 μm nylon mesh to generate cardiac cell suspensions. Cells were then counted and used for flow cytometry, or real-time PCR.

*Competitive repopulation assays*: Competitive repopulation assays were conducted as described previously [2]. Briefly, 2.103 Lin−/Sca-1+/c-Kit+ cells (HSC) sorted from the AT or the BM of donor mice were mixed with 2.105 competitor BM total cells from C57Bl/6 mice. In all the experiments, control and experimental HSC were sorted from animals of equal age. The mixed population was intravenously injected into lethally irradiated (10Gy, 137Cs source) recipient mice of equal age, previously anesthetized by inhalation of isoflurane (2.5%). Mice reconstituted with AT- or BM-HSC (AT-or BM-mice) were then allowed to recover for 8 weeks. All experiments were carried out in compliance with European Community Guidelines (2010/63/UE) and approved by the Institutional Ethics Committee (US 006/CREFRE; CEEA-122) and from the Ministry of National Education, Higher Education and Research (protocol reference: 10691). Chimerism was assessed by quantifying Tomato^+^ (dT^+^) cells among total CD45^+^ cells in the SVF.

*Acute myocardial infarction*: Myocardial infarction was induced by left coronary ligation as previously described [31,32]. All experiments were conducted according to the Ethical Committee for Animal Experimentation (University of Paris, CEEA 34) and the National Charter on the Ethics on Animal Experimentation from the French Minister of Higher Education and Research under the reference MESR: n° 01373.01. Briefly, after their anesthesia using ketamine (100 mg/kg body weight) and xylazine (10 mg/kg body weight), mice underwent a thoracotomy in the fourth left intercostal space, and a ligation of the left anterior descending artery. The same procedure was performed on the sham-operated mice except that the ligation was not tied. Transthoracic echocardiography was performed using an echocardiograph (ACUSON S3000 ultrasound) equipped with a 14 MHz linear transducer (1415SP) as previously described [31,32]. Two-dimensional parasternal long-axis views of the left ventricle (LV) were obtained for guided B-mode measurements at the end of the diastole (d) and systole (s). For functional analysis of the LV viable myocardium, we evaluated the LV end diastolic and end systolic volume. Percentage ejection fraction (%EF) was calculated using the following formula: %EF = [(LVEDV-LVESV)/LVEDV] × 100 [32]. For evaluation of cardiac remodeling, hearts were excised, rinsed in PBS and frozen in liquid nitrogen. Hearts were cut by a cryostat (CM 3050S, Leica, Paris, France) into 5–7 μm thick sections. Masson’s trichrome and Sirius red staining were performed for infarct size and myocardial fibrosis evaluation, respectively. Infarct size was calculated as the ratio of total infarct circumference divided by total left ventricle circumference, as previously described [31,32]. The collagen volume fraction was calculated as the ratio of the total area of interstitial fibrosis to the myocyte area in the entire visual field of the section. Endothelial cells within capillaries were visualized after Bandeiraea Simplicifolia lectin staining (1:100, FITC-conjugated Griffonia simplicifolia, Sigma-Aldrich, Saint-quentin Fallavier, France), and cardiomyocytes after wheat germ agglutinin staining (1:200, Texas Red-conjugated, ThermoFisher, Montigny le Bretonneux, France). The results are expressed as a ratio of the number of capillaries per cardiomyocytes.

*Flow cytometry analysis and cell sorting*: Single-cell suspensions of mice spleen, BM, heart, blood and SVF were stained in PBS containing FcR-blocking reagent for 30 min at 4 °C with fluorescent-conjugated antibodies. Phenotyping was performed by immunostaining with conjugated rat anti-mouse Abs and compared with isotype-matched control Abs (Appendix A). Events were then acquired on a LSR Fortessa flow cytometer (BD Biosciences, Le pont de claix, France). Data acquisition and analysis were performed using Diva (version 8, Becton Dickinson, San Jose, CA, USA) and Kalusa version 1.2 (Beckman Coulter, Villepinte, France) or FlowJo software version 10.4.2 (FlowJo LLC, Becton Dickinson, Ashland, OR, USA ). Dead cells, debris or red blood cells were excluded according to forward scatter (FSC) and side scatter (SSC) profiles. CD45^+^ leukocytes were then gated: dendritic cells were identified as CD11c^high^MHCII^high^ cells, neutrophils as CD11b^+^Ly6G^+^ cells, monocytes as CD11b^+^F4/80^−^Ly6G^-^Ly6C^high^ or Ly6C^low^ cells, macrophages as CD11b^+^Ly6G^-^CD64^+^F4/80^+^ cells. CD11b^-^ leukocytes were recognized and then gated on B220 and CD3: B lymphocytes were characterized as B220^+^ cells, T cells as CD3^+^ cells and then stratified by CD4 and CD8 expression.

For HSC-sorting experiments, SVF cells were stained with Ly-6A/E (Sca-1), CD117 (c-Kit) antibodies, and Lineage Panel (Lin). Cells negative for lineage markers were gated, and Sca-1 and CD117 double-positive cells were sorted (BD FACSAria Fusion III Cell sorter, BD Biosciences), as previously described [2]. Enrichment of the HSC was determined by flow cytometry and varied between 92% and 97%.

For cardiac macrophage sorting, hearts were collected and enzymatically digested for 30 min in a buffer containing Collagenase II (1 mg/mL, Serlabo technologies, Entraigues, France), Collagenase IV (1 mg/mL, Serlabo technologies, Entraigues, France ) and Protease XIV (0.1 mg/mL, Sigma-Aldrich). Single-cell suspension was then obtained by passing through a 40 μm cell strainer, before staining with fluorescently tagged antibodies and eFluor780-conjugated Fixable Viability Dye (eBioscience, Villebon sur Yvette, France).

*Characterization of cardiac macrophages*: FACS-sorted cardiac macrophages were then used to assess cytokine levels using Bio-Plex Multiplex Immunoassay System (Bio-Rad, Marnes la coquette, France).

*Statistical analysis*: Variance between three or more independent groups was compared by using Kruskal–Wallis one-way ANOVA. Statistical differences were measured using Dunn’s multiple comparisons test or non-parametric Mann–Whitney Test. All statistical analyses were carried out using GraphPad Prism 9.0 software (Ritme, Paris, France). A *p* value < 0.05 was considered significant. The following symbols for statistical significance were used throughout the manuscript: * *p* < 0.05; ** *p* < 0.01; *** *p* < 0.001.

## Figures and Tables

**Figure 1 ijms-23-10498-f001:**
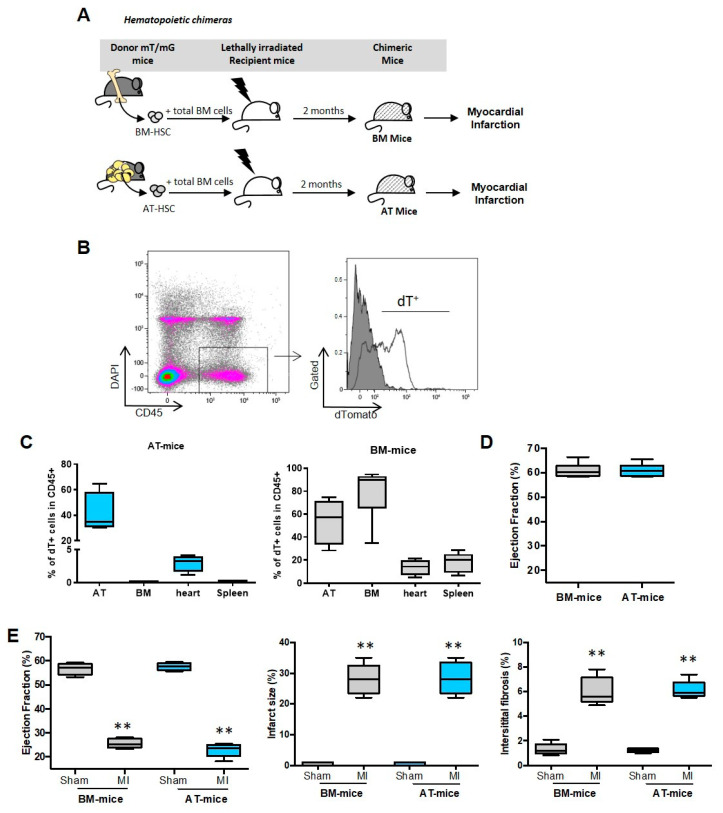
Establishment of chimeric mice models to reveal the distinct role of adipose tissue or bone marrow-derived inflammatory cells. (**A**) Hematopoietic chimera strategy: 2.10^3^ hematopoietic stem cells (HSC) sorted from sub-cutaneous adipose tissue (AT) or bone marrow (BM) of mTmG mice were co-injected with 2.10^5^ total BM cells from C57Bl6 mice into lethally irradiated C57Bl6 recipients. Eight weeks after hematopoietic reconstitution, chimeric mice were subjected to myocardial infarction. (**B**) Representative dot plots of flow cytometry analyses showing the chimerism (tdT^+^ cells among CD45^+^ cells; open histogram) or wild type cells (dT^-^; grey histogram) in the AT of AT- chimeric mice (n = 4). (**C**) Quantification of the chimerism in sub-cutaneous AT, BM, heart and spleen of AT- and BM-chimeric mice (n = 4–6). (**D**) Ejection Fraction in unoperated AT- and BM chimeric mice (n = 8). (**E**) Quantification of ejection fraction, infarct size and interstitial fibrosis in sham and infarcted chimeric mice (n = 5). Comparisons between 2 group were performed using Dunn’s multiple comparisons test. ** *p* < 0.01.

**Figure 2 ijms-23-10498-f002:**
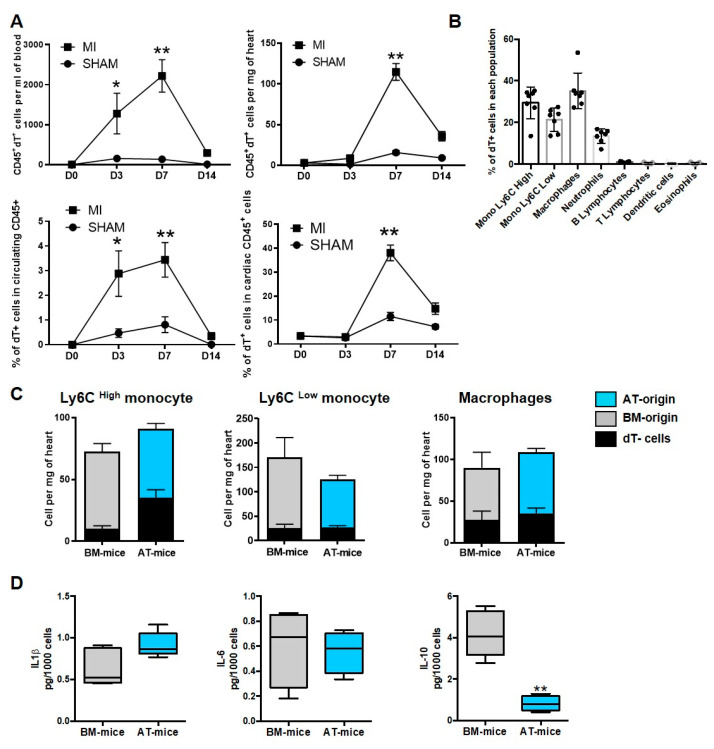
Adipose tissue is a reservoir of monocytes and macrophages populating the infarcted heart. (**A**) Quantification of CD45^+^/dT^+^ in blood and heart of sham and infarcted AT-chimeric mice (n = 4–9). (**B**) Percentage of dT^+^ cells in myeloid and lymphoid populations, in the heart of AT-mice, 7 days post MI (n = 7). (**C**) Quantification of dT^+^ and dT^-^ Ly6C^high^, Ly6C^low^ monocytes and macrophages in the heart, 7 days after MI in BM- and AT-chimeric mice (n = 4–10). In the stacked bars, black columns indicate the number of dT^-^ cells and colored columns the number of dT^+^ cells in BM-mice (grey) or AT-mice (blue). (**D**) Quantification of IL-1α, IL-6 and IL-10 production in cardiac macrophages sorted from BM- and AT-mice 7 days post MI (n = 4–6). Comparisons between groups were performed using Dunn’s multiple comparisons test (B) or Mann-Whitney Test (**D**). * *p* < 0.05; ** *p* < 0.01.

**Figure 3 ijms-23-10498-f003:**
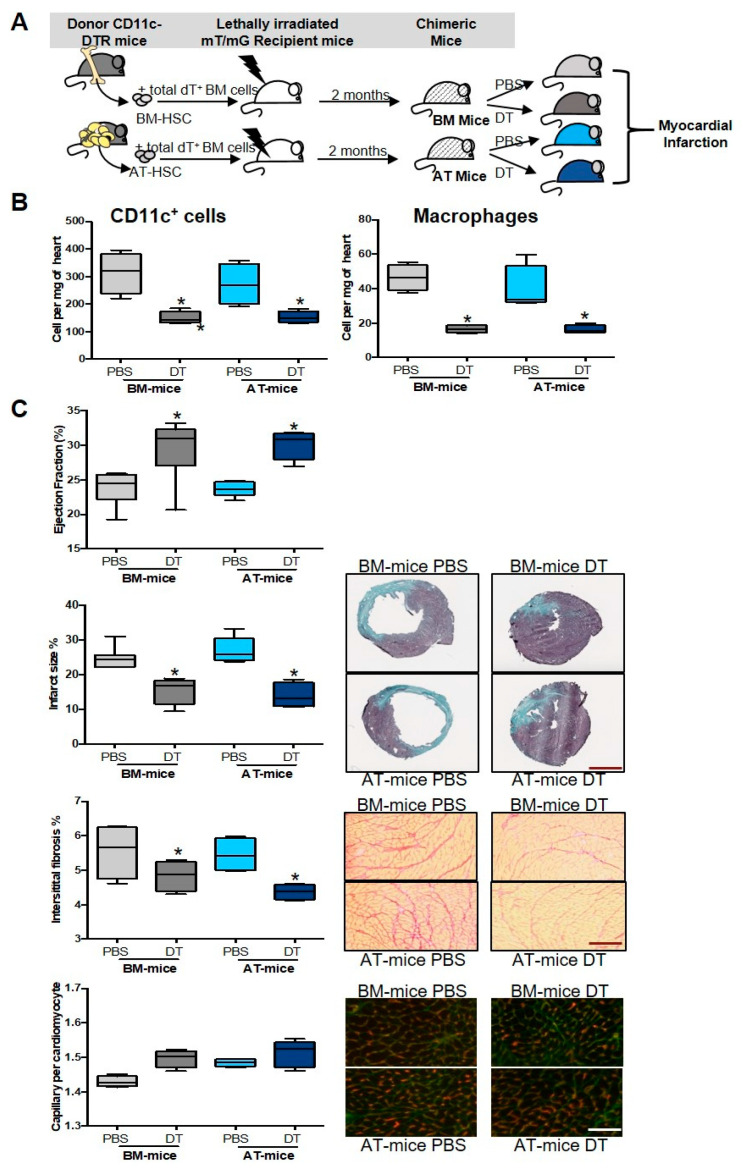
Depletion of adipose tissue-derived CD11c^+^ macrophages improves cardiac remodeling and function after acute MI. (**A**) Schematic representation of experimental procedure: lethally irradiated mT/mG recipient mice were co-injected with hematopoietic stem cells (HSC) sorted from sub-cutaneous adipose tissue (AT) or bone marrow (BM) and total BM cells isolated from CD11c-DTR and mT/mG donor mice, respectively. Eight weeks after reconstitution, chimeric mice were treated with diphtheria toxin (DT) or PBS as control at day 0, 3 and 6 after the ischemic insult. (**B**) Quantification of CD11c^+^ cells and CD11c^+^ macrophages in the heart, 7 days post-MI in BM- and AT-mice (n = 3–4). (**C**) Representative photomicrographs and quantification of ejection fraction, infarction size (scale bar, 2 mm), interstitial fibrosis (scale bar, 10 µm) and capillary density (scale bar, 10µm) in BM and AT-mice treated with or without DT (n = 4–8). Comparisons between groups were performed using Dunn’s multiple comparisons test. * *p* < 0.05 in DT vs. PBS treated mice.

**Figure 4 ijms-23-10498-f004:**
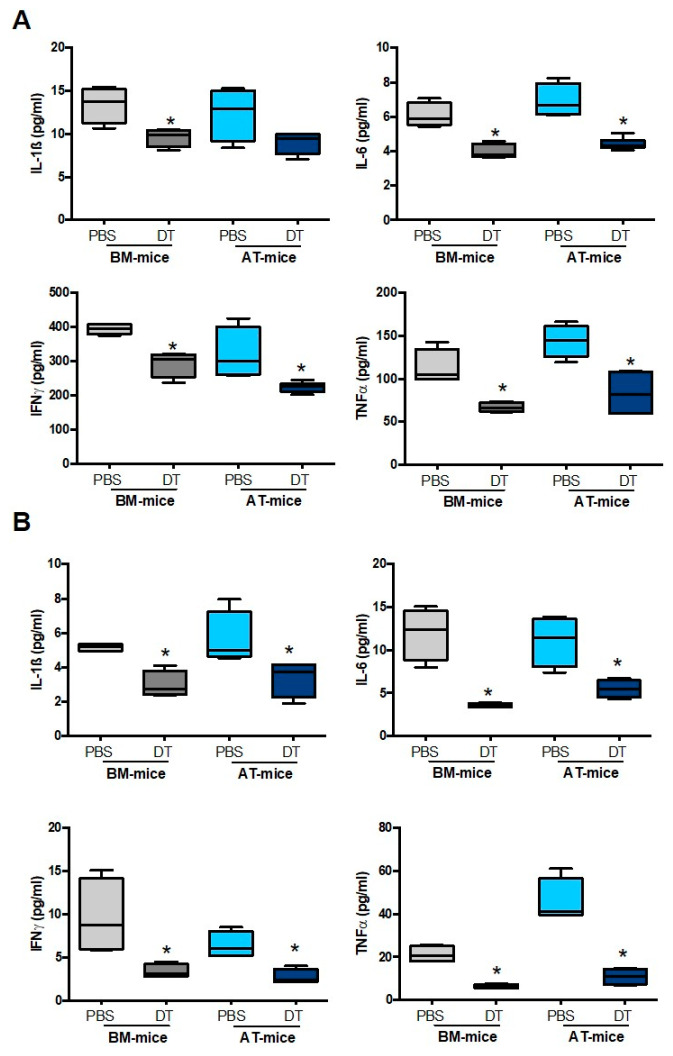
Depletion of adipose tissue-derived CD11c^+^ macrophages lowers cardiac inflammation. IL-1α, IL-6, IFNγ and TNFα levels in the heart (**A**) and the blood (**B**) of AT- and BM- chimeric mice treated with or without diphtheria toxin (DT), 7 days post- MI (n = 4–6). Comparisons between groups were performed using Dunn’s multiple comparisons test. * *p* < 0.05 in DT vs. PBS treated mice.

**Figure 5 ijms-23-10498-f005:**
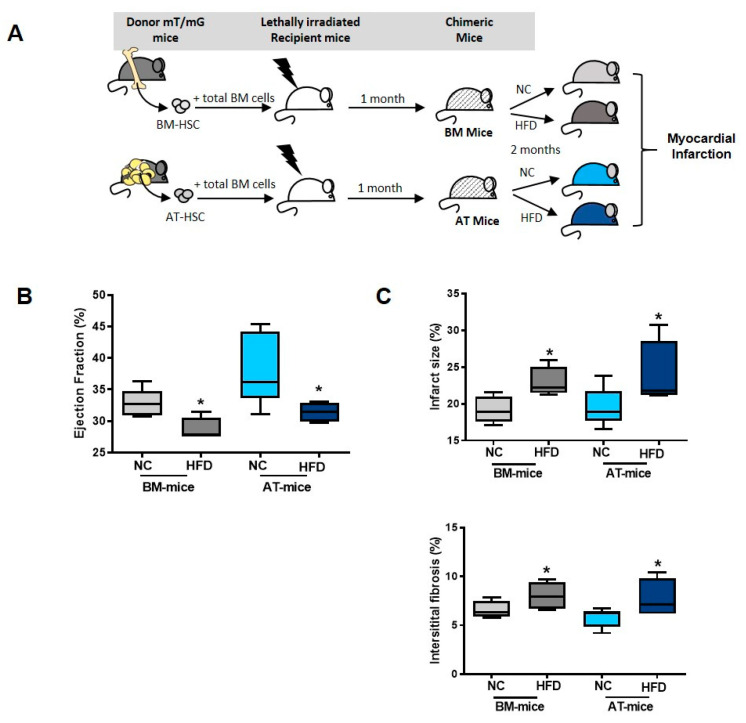
High Fat Diet worsens cardiac remodeling and function. (**A**) Schematic representation of experimental procedure: lethally irradiated recipient mice were co-injected with hematopoietic stem cells (HSC) sorted from sub-cutaneous adipose tissue (AT) or bone marrow (BM) and total BM cells isolated from mT/mG donor and C57Bl6 donor mice, respectively. Eight weeks after reconstitution, mice were then fed a normal Chow (NC) or a high fat diet (HFD) for 2 months. (**B**) Ejection fraction and (**C**) infarct size as well as interstitial fibrosis in BM- and AT- mice fed a normal Chow (NC) or a high fat diet (HFD) for 2 months and then challenged with MI (n = 4–10). Comparisons between groups were performed using Dunn’s multiple comparisons test. * *p* < 0.05 in HFD vs. NC fed mice.

**Figure 6 ijms-23-10498-f006:**
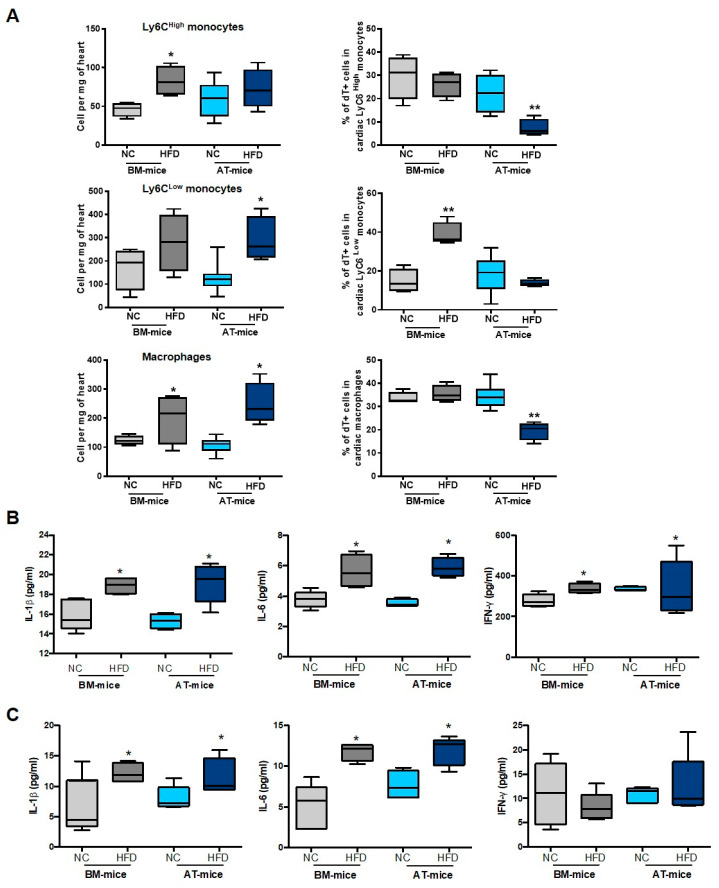
Diabetic chimeric mice exhibit higher cardiac inflammation. (**A**) Quantification of total and dT^+^ Ly6C^High^and Ly6C^Low^ monocytes, and macrophages in the heart of BM- and AT-mice fed a normal chow (NC) or a high fat diet (HFD) for 8 weeks (n = 4–8). (**B**,**C**) Cytokine levels in the heart (**B**) and the blood (**C**) of BM- and AT-mice fed a normal chow (NC) or a high fat diet (HFD) for 8 weeks (n = 4–5). Analyses have been performed 7 days after the onset of acute MI. Comparisons between groups were performed using Dunn’s multiple comparisons test. * *p* < 0.05 and ** *p* < 0.01 in HFD vs. NC fed mice.

**Figure 7 ijms-23-10498-f007:**
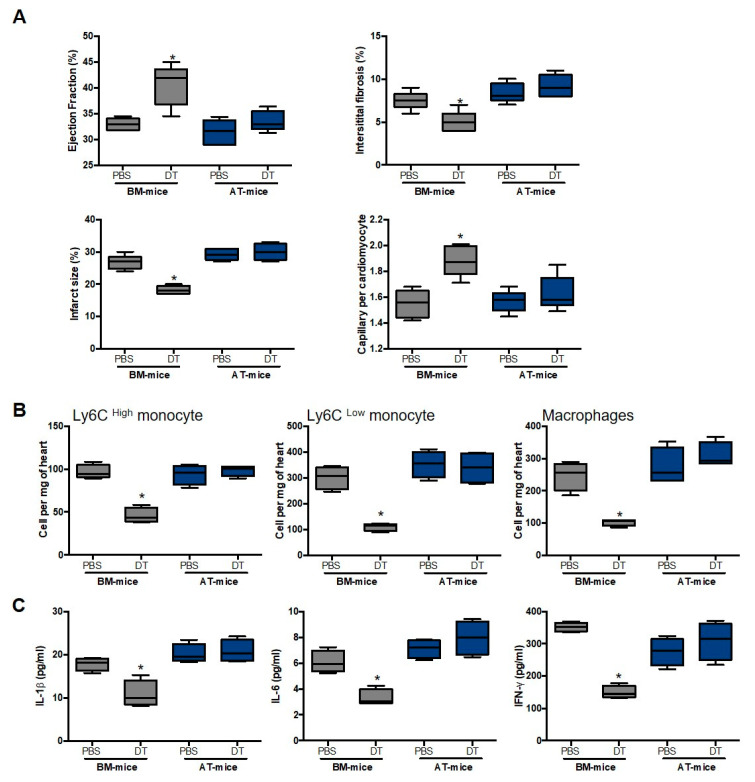
Adipose tissue-derived CD11c^+^ macrophage depletion did not regulate cardiac function and remodeling in diabetic mice. (**A**) Ejection fraction, interstitial fibrosis, infarct size and capillary density in diabetic BM- and AT-chimeric mice treated with or without diphtheria toxin (DT) (n = 6–8). (**B**) Monocyte and macrophage number in the heart of diabetic BM- and AT-chimeric mice treated with or without DT, (n = 6–8). (**C**) IL-1α, IL-6 and IFNγ levels in the heart of diabetic BM- and AT-chimeric mice treated with or without diphtheria toxin (DT) (n = 4–6). Comparisons between groups were performed using Dunn’s multiple comparisons test (* *p* < 0.05 in DT vs. PBS treated mice).

## Data Availability

The datasets used and/or analyzed during the current study are available from the corresponding author on reasonable request.

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
