# Peer review of "Identification of Adipose Tissue as a Reservoir of Macrophages after Acute Myocardial Infarction"

_ijms, 2022, doi:10.3390/ijms231810498_

Round 1

Reviewer 1 Report

The article is devoted to the current direction - the study of the characteristics of adipose tissue macrophages in myocardial infarction and diabetes mellitus. The introduction needs to be revised, since the authors do not convincingly show the need for a study. Unfortunately, I must admit that I was mistaken I accepted the invitation to act as a reviewer. I cannot qualitatively evaluate the performed experiment. My competencies are more in line with clinical pathophysiology. However, I would like to note that the work is relevant. However, a re-review is needed. I ask the authors to once again revise the article for clarity of presentation of thoughts, translation errors, and the level of English. I apologize to the authors and editor for my mistake.

Author Response

We would like to thank the reviewer for her/his comments. The lack of clarity in the introduction was probably due to the fact that the first part contained a piece of text describing the instructions to authors that was unfortunately inserted during the formatting of the manuscript during submission. We apologize for this error. The text was also proofread by an English native.

Reviewer 2 Report

The main aim of study of Gomez et al. was to characterize an impact of repopulation of macrophages in irradiated mice either from bone marrow or subcutaneous adipose tissue on myocardial infarction. The study fucuses on a very important topic of organ crosstalk and the conclusions of the work are thoroughly supported by experiments. I have only minor issues:

·         At the beginning of Introduction part of another text probably got inadvertently copied.

·         I missed citation for the statement in Introduction „…Diabetic patients displayed imbalanced inflammatory versus reparative macrophages ratio….“

·         Please check for inconsistencies between Results, Fig 1A and Fig 1 description in the length of regenerative phase after hematopoietic reconstitution.

·         The „gating strategy“ in Fig 2A is not properly described. Please include gating strategy for all cell types into supplemental information.

·         Illustrative images in Fig 3 and 5 are so small that it is not possible to see any details – please enlarge it.

·         Please add into Materials and Methods description of evaluation of capillary density.

·         Please discuss possible differences in mouse vs. human inflammatory response and the probable impact of differences in macrophages on myocardial infartion in humans.

Round 2

Reviewer 1 Report

As I noted earlier, the article is undoubtedly relevant. The corrections which  are made, most likely initiated by a molecular biologist, made the article even more qualitative and attractive to readers.